# Construction of a Real-Time Ship Trajectory Prediction Model Based on Ship Automatic Identification System Data

**Daping Xi [1], Yuhao Feng [2], Wenping Jiang [3], Nai Yang [1], Xini Hu [1,*] and Chuyuan Wang [1]**

[1] School of Geography and Information Engineering, China University of Geosciences, Wuhan 430000, China; xidp@cug.edu.cn (D.X.); yangnai@cug.edu.cn (N.Y.); wcy1998@cug.edu.cn (C.W.)
[2] Zhejiang Institute of Communications Co., Ltd., Hangzhou 310000, China; fengyh@zjic.com
[3] School of Resource and Environmental Sciences, Wuhan University, Wuhan 430000, China; andyjiang@whu.edu.cn
[*] Correspondence: cug_sydneyhu@cug.edu.cn

**Abstract:** The extraction of ship behavior patterns from Automatic Identification System (AIS) data and the subsequent prediction of travel routes play crucial roles in mitigating the risk of ship accidents. This study focuses on the Wuhan section of the dendritic river system in the middle reaches of the Yangtze River and the partial reticulated river system in the northern part of the Zhejiang Province as its primary investigation areas. Considering the structure and attributes of AIS data, we introduce a novel algorithm known as the Combination of DBSCAN and DTW (CDDTW) to identify regional navigation characteristics of ships. Subsequently, we develop a real-time ship trajectory prediction model (RSTPM) to facilitate real-time ship trajectory predictions. Experimental tests on two distinct types of river sections are conducted to assess the model's reliability. The results indicate that the RSTPM exhibits superior prediction accuracy when compared to conventional trajectory prediction models, achieving an approximate 20 m prediction accuracy for ship trajectories on inland waterways. This showcases the advancements made by this model.

**Keywords:** AIS data; river sinuosity; trajectory clustering; anastomosing river; trajectory prediction

## 1. Introduction

In contrast to land transport, water transport lacks well-defined fixed routes, granting ships a greater degree of freedom in their movement. Consequently, the management of water transport is inherently more intricate. Ship collisions are an unfortunate occurrence during waterborne navigation, often stemming from factors like equipment malfunctions and human errors. Thus, real-time monitoring of the ship's course becomes indispensable for timely detection of anomalous behavior and the reduction of the risk of ship accidents. To support the safety of ship navigation, the water traffic management authorities employ a suite of modern information equipment, including the Automatic Identification System (AIS) and Global Positioning System (GPS) [1].

AIS allows real-time monitoring of vessels within the base station's coverage area by receiving and processing information transmitted by shipboard AIS equipment, leveraging unique AIS equipment codes for automatic ship identification. While AIS data discloses the location of the most recent AIS message sent, it carries inherent transmission delays. Additionally, there's a time lag associated with data transmission, parsing, loading, and display, hindering real-time ship position representation. Moreover, equipment failures and signal interference can lead to the loss of AIS data. These challenges in the domain of AIS data complicate the task of ensuring ship safety.

To address these issues, we present a real-time ship trajectory prediction model (RSTPM) designed to enable real-time ship trajectory monitoring and prompt detection of irregular behavior. This model offers valuable applications in ensuring the safety of ship navigation.

## 2. Related Work

According to the relevant research based on AIS data at home and abroad, we summarize both the advantages and disadvantages of various ship trajectory prediction methods, as indicated in Table 1.

**Table 1.** Classification of trajectory prediction methods and brief description of advantages and disadvantages.

| Categories | Method Categories | Advantages | Disadvantages |
|---|---|---|---|
| Simulation Method | Exponential smoothing model (ESM) [2] | Predictions can be made with a small amount of data | Only short-term forecasts can be made |
| | Curvature velocity method [3] | Simple model and good real-time | Only short-term forecasts can be made |
| Statistical Methods | Kalman filter [4] | Linear, unbiased, high accuracy | Relies on raw data and cannot predict over time |
| | Autoregressive moving average model (ARIMA) [5] | Simple model and wide application | Requires large amounts of data and low accuracy |
| | Hidden Markov model (HMM) [6] | Good state prediction of the process | Poor robustness and complex parameter settings |
| | Gaussian mixture model (GMM) [7] | High accuracy in short range prediction | Vulnerable to data complexity and low utility |
| | Bayesian Networks [8] | Efficient and easy to train | Vulnerable to prior probabilities and input variables |
| Machine Learning | K-nearest neighbor (KNN) [9] | Easy to implement, no parameter estimation required | Accuracy suffers when sample size is unbalanced |
| | Support vector machine (SVM) [10,11] | Applicable to linear and nonlinear problems | Only for dichotomous problems |
| | Artificial neural network (ANN) [12] | High accuracy and error tolerance to noise | Requires large number of initial parameters and long training time |
| | Extreme learning machine (ELM) [13] | No iterations for hidden layers, fast learning | May cause overfitting problems |
| | Backpropagation (BP) [14,15] | Ability to learn and generalize on your own | May fall into local extremes leading to training failure |
| Deep Learning | Long short-term memory (LSTM) [16–19] | The deficiency of long-term dependence in recurrent neural network (RNN) is effectively improved. | The internal structure is relatively complex and time-consuming to calculate |
| | GRU [20] | Simple model, better training speed than LSTM | Cannot completely solve the gradient disappearance problem |
| | GAN [21] | Can produce clearer and more realistic samples | Not suitable for handling discrete data, such as text |
| | Convolutional neural network (CNN) [22] | Feature extraction can be performed automatically | Training results easily converge to local minima |
| | Deep neural network (DNN) [23] | Very good nonlinear fitting ability | Difficult to train, requires a large amount of data |
| Other | Hybrid model [24–27] | Combines the advantages of multiple models | May result in an increase in calculated costs |

### 2.1. Ship Trajectory Prediction Based on Simulation Methods

Simulation methods involve creating physical models to simulate real ship behavior. This method is rarely used alone in ship trajectory prediction; it is generally combined with other methods to form a hybrid method for prediction.

The ESM is used to predict the location, course, and speed of the ship; meanwhile, the actual collision scene of the ship is analyzed. This method has been shown to achieve the prediction of ships' behavior [2]. Mazzarella proposed a Bayesian algorithm based on particle filters that uses KNN to match the current trajectory sequence of the ship, enabling the prediction of ship trajectories when traffic route data are available [3].

## 2.2. Ship Trajectory Prediction Based on Statistical Methods

The statistical-based approach assumes that the historical trajectory of a ship and the predicted trajectory have a certain similarity, and the prediction is achieved by fitting the ship trajectory.

Ju et al. [4] proposed a multi-layer architecture interactive-aware Kalman neural network to solve the problem of mutual interaction in the transportation system. A differential ARIMA model was used to predict ship trajectories from ship AIS data, which is applicable to the detection of ship collision avoidance [5]. The trajectory sequence is transformed into column vectors through wavelet transform, which are then used as the input for HMM. This is an algorithm (HMM-WA) that increases the accuracy of ship trajectory prediction [6]. Rong et al. [8] presented a model based on Gaussian processes and uncertain acceleration, designed to achieve real-time monitoring of ships during navigation.

## 2.3. Ship Trajectory Prediction Based on Machine Learning

Unsupervised learning mainly focuses on the clustering and dimensionality reduction of data, while supervised learning has a broader range of applications. For example, KNN and SVM can predict ship trajectories by learning the motion characteristics of ship trajectories.

Duca et al. [9] proposed a model for trajectory prediction based on a KNN classifier, considering five characteristics of ships: longitude, latitude, heading, speed, and type. The model's prediction accuracy was verified. Liu et al. [10] developed an online multioutput model based on a selection mechanism. The model can achieve high prediction accuracy with small samples. Additionally, an SVR-based trajectory prediction model was proposed, but the sample data and parameters required for the model cannot be changed during model training [11]. Gan et al. [12] used the clustered ship trajectory and other known factors, such as ship speed, to establish an ANN model for predicting the ship's trajectory.

Since the advent of deep learning, it has demonstrated excellent performance in many tasks, including the prediction of ship trajectories. The advantage of LSTM over BP neural networks lies in its ability to process and analyze time series and sequence data. The gate structure in LSTM's internal network enables it to mine trends and correlations in sequence data, resulting in a better prediction effect when applied to time series data, such as traffic and location. Moreover, LSTM's prediction accuracy is better than that of BP neural networks [16–18,28], making it applicable to long-term prediction. Gao et al. [29] studied a bi-directional LSTM (Bi-LSTM) network, aiming to enhance the memory ability of historical data and the correlation between future time series data. Liu et al. [19] integrated convolutional transformations into a Bi-LSTM based on an attention mechanism in order to achieve long-term prediction. In another study, ship trajectory sequence features extracted by CNN are input into the LSTM model for prediction [22]. In addition to LSTM, GRU, CNN, and GAN, there are also DNN-based frameworks for predicting the trajectory of merchant ships (such as tankers and container ships). However, the DNN module is prone to overfitting and may not achieve high accuracy during training [23].

## 2.4. Ship Trajectory Prediction Based on the Hybrid Method

The hybrid method focuses on combining the advantages of various methods to enhance the efficiency of trajectory prediction tasks. Murray & Perera [24] proposed an algorithm that combines GMM, KNN, and bilinear automatic coding to build the prediction model. Schöller et al. [26] first used kernel density estimation to convert historical AIS data into probabilistic heat maps and then used a convolutional autoencoder for further coding. They constructed a model based on GAN and LSTM to achieve ship trajectory prediction. Additionally, Suo et al. [27] constructed a hybrid model based on DBSCAN and GRU to achieve real-time prediction.

Moreover, the clustering methods of ship trajectory are roughly divided into distance-based [30–33], density-based [34–36], and statistical-based [37,38] methods. Most studies have achieved efficient clustering of ship trajectories by combining the advantages of

various clustering methods [30,31,35–37,39]. In this study, a new approach to cluster ship trajectory, called the combination of DBSCAN and DTW (CDDTW), is proposed, which combines the optimized DBSCAN algorithm (based on density) and the improved DTW algorithm (based on distance) to cluster ship historical trajectories. The proposed method also extracts the regional navigation characteristics of ship trajectory based on the clustering results. An RSTPM based on an LSTM is constructed.

## 3. Method

To enhance the driving security of inland waterway ships, the RPSTM is proposed to provide real-time prediction of ship positions. We conducted tests on two types of river sections to verify the reliability of the RPSTM. Figure 1 shows the technical road plan for the study.

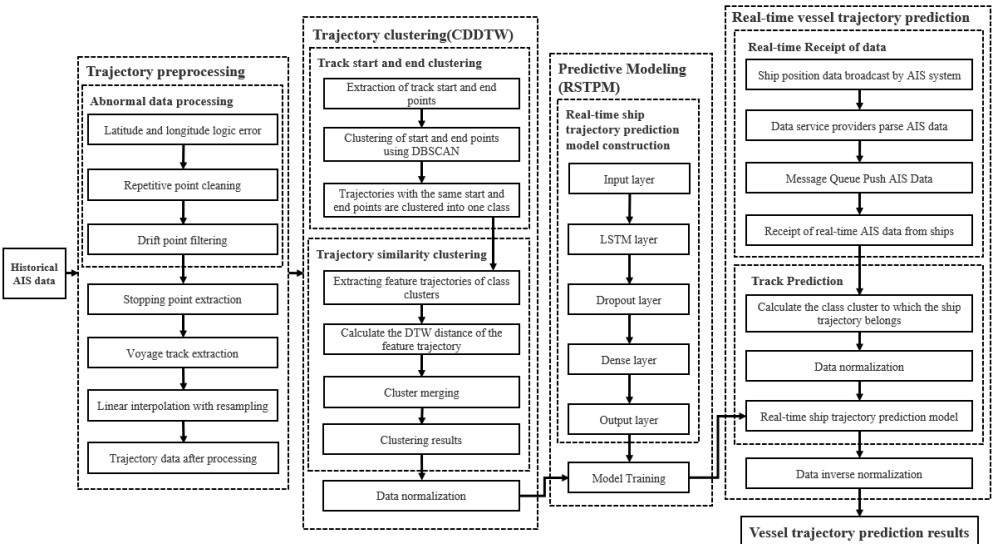

**Figure 1.** Technical roadmap for the RSTPM.

### 3.1. AIS Data Preprocessing

AIS data is a type of spatiotemporal trajectory data that records the location, navigation status, and other behavioral characteristics of the ship. During the reception of the original AIS data, some of the AIS data may be lost due to communication signal issues. Therefore, we preprocess the AIS data to eliminate redundancy and improve data quality. The steps for preprocessing the AIS trajectory data are as follows: (1) Filtering the noise data. The dynamic information in the AIS data is automatically obtained through the GPS module. Due to weather, air, and signal interference, there will be noise data such as position anomaly and heading anomaly in the data broadcasted by the AIS equipment. (2) Detecting the stay point of the ship. The position data of ships is constantly generated during the navigation process; however, the data generated when the ship is anchored is unnecessary for the prediction of navigation. Therefore, we extract and delete these data. For example, if the ship stays in a place for more than 30 min, we filter out the location data for that time period. (3) Extracting the ship's navigation trajectory from AIS data. (4) Using two approaches to solve the missing data. The static data can be checked with the ship database of the management department. For the dynamic data, such as position, speed, etc., we use linear interpolation and re-sampling to make the positioning time interval of the adjacent two points in the trajectory consistent. Under this processing, the trajectory data become denser. (5) Normalizing the data to ensure that the subsequent neural network training has lower complexity and faster solution speed. The variation of ship AIS data volume in different areas is shown in Table 2.

**Table 2.** Variation of ship AIS data volume in different areas.

| Study Area | Number of Raw Data | Number of AIS Data after Step (1) Processing | Number of AIS Data after Step (4) Processing |
|---|---|---|---|
| The Wuhan section of the dendritic river system in the Yangtze River's middle reaches | 8,538,423 | 1,767,305 | 4,428,680 |
| The partial reticulated river system in the northern part of the Zhejiang Province | 1,850,761 | 217,770 | 1,302,994 |

### *3.2. Ship Trajectory Clustering*

In this part, we optimize the DBSCAN and DTW algorithms for better ship trajectory clustering. Aiming at the problem of slow neighborhood query in the DBSCAN algorithm, the Ball-Tree algorithm is introduced to optimize the query efficiency. Meanwhile, a cross-point matching method is proposed to optimize the matching path to solve the multi-point matching problem in the DTW algorithm. The workflow of the CDDTW algorithm is as follows: First, the optimized DBSCAN is used to cluster the navigation trajectory through the ship's starting points and ending points. Second, the improved DTW algorithm is used to calculate the trajectory similarity distance in each cluster. In this step, trajectories with large similarity distance will be reclassified into different clusters This approach can effectively detect situations where the starting point and ending point of the trajectory are the same, while the route is quite different, thereby improving the clustering results. Third, the feature trajectory is extracted from each cluster (the feature trajectory: the trajectory with the smallest total distance between itself and all other trajectories in the cluster). Finally, we use the improved DTW algorithm to obtain the similarity distance between the feature trajectories. This step aims to avoid the situation where similar trajectories are divided into different clusters due to the opposite starting points and end points. After these steps, we get a final set of different feature trajectories.

### 3.2.1. The CDDTW Trajectory Clustering Algorithm

We introduce the Ball-Tree algorithm into the DBSCAN algorithm to accelerate the query speed of neighborhood points. In the process, a segmentation threshold N is introduced to construct the Ball-Tree, which stops the segmentation of the node when the number of samples contained in the subtree node is less than or equal to N. The segmentation outcomes of the samples for different values of N are shown in Figure 2. The comparison of calculation time before and after algorithm optimization is shown in Table 3.

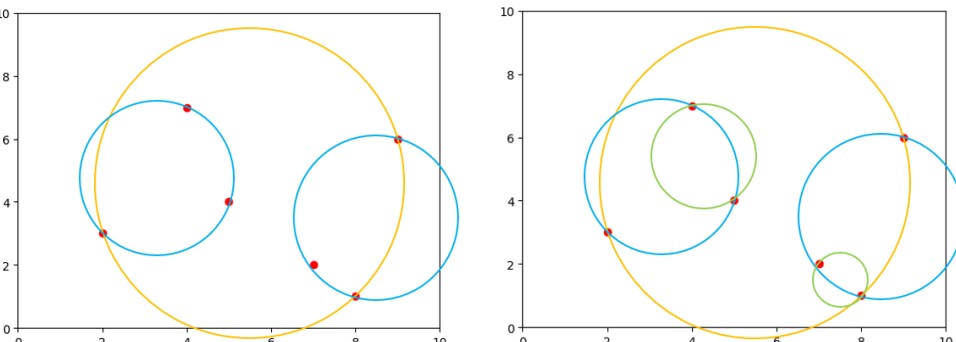

**Figure 2.** Segmentation results of Ball-Tree with different segmentation thresholds (*N* = 3, **left**) (*N* = 2, **right**).

**Table 3.** The clustering time comparison of the algorithm under different numbers of sample points (unit: s).

| Algorithm | Number of Sampling Points | | | | |
|---|---|---|---|---|---|
| | **100** | **500** | **1000** | **3000** | **5000** |
| DBSCAN | 0.205 | 0.940 | 3.352 | 28.84 | 81.8 |
| DBSCAN + Ball-Tree | 0.142 | 0.201 | 0.298 | 0.801 | 1.533 |

The clustering results of the DBSCAN algorithm are mainly affected by two parameters: one is the scanning radius of the neighborhood E, called R, and the other is the minimum number of points in the neighborhood E, called MinP. The values of R and MinP interact with each other. This is because, with the increase of R, the surrounding noise points will be continuously absorbed into the cluster, causing the number of noise points to gradually decrease. Consequently, the number of clusters will also decrease synchronously. In the above process, the noise and clusters will first decrease sharply, then decrease steadily, and finally tend to level off. Therefore, there will be a transitional region in the process. This interval is the best value range of MinP and R.

Taking the AIS data of the partial reticular river system in Zhejiang as an example, with the change of MinP and R, the variation of noise points and the number of clusters are shown in Figure 3. The chart shows that when MinP is 3, 4 or 5, the number of clusters is smaller than in other situations, and there is an obvious transitional region However, when MinP is 6, the number of clusters rebounds with the increase of R. In our ship trajectory prediction task, we should cluster similar trajectories as much as possible to reduce the number of clusters and thus increase the training data in each cluster for the prediction model. Therefore, we set the MinP to 5 and the R to 0.001.

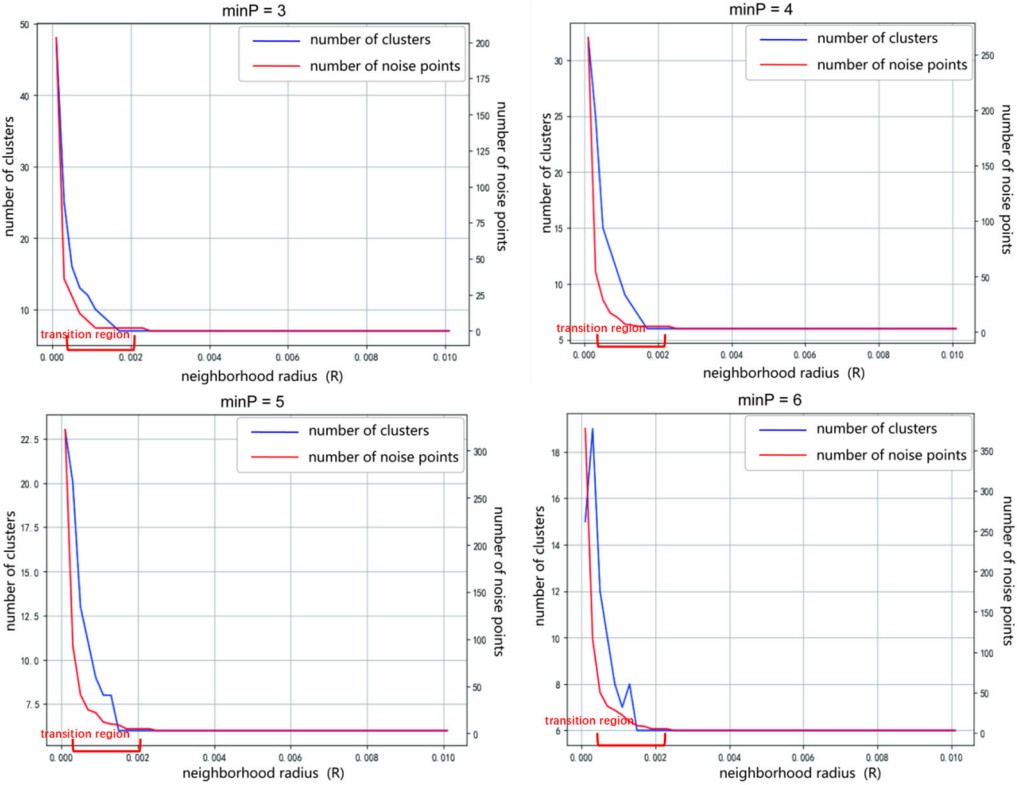

**Figure 3.** The influence of two clustering parameters on the number of clusters and the number of noise points.

The DTW algorithm [40] is a classical time series distance measurement method. When using DTW to calculate the shortest distance, it is necessary to form a distance matrix of two time series, and then accumulate the elements passing from the lower left corner to the upper right corner of the distance matrix to minimize the sum of the accumulated distances. Since the DTW algorithm requires monotonicity when matching, this means that any point in the two sequences cannot be skipped. It will cause one point in a sequence to need to match multiple points in another sequence, posing a problem in reducing the similarity between two time series. To solve this problem, we propose a cross-point matching method to avoid the situation of one point matching multiple points, while ensuring that the matching path aligns with the diagonal as much as possible, as shown in Figure 4. Here, the DTW distance between the two trajectory sequences is shown in Formula (1), where M and N are the lengths of the two sequence points, and dis (m, n) is the Euclidean distance between the sequence point m and the sequence point n.

$$DTW(m,n) = \begin{cases} 0 & M = 0 \ and \ N = 0 \\ \infty & M = 0 \ or \ N = 0 \\ dis(m,n) + min \begin{cases} DTW(m-1,n-1) \\ DTW(m-2,n-1) \\ DTW(m-1,n-2) \end{cases} & otherwise \end{cases} \tag{1}$$

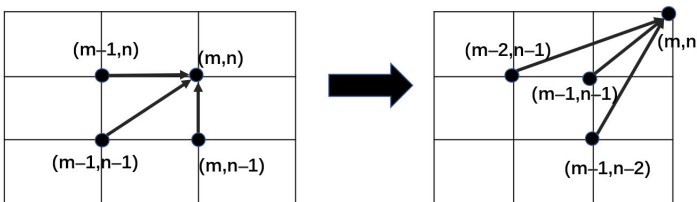

**Figure 4.** Improvement of the algorithm.

Figure 5 shows two different ship navigation trajectories. The original DTW algorithm and the improved DTW algorithm are used to calculate the similarity distance matrix grayscale between the two trajectories. The similarity distance matrix grayscale and matching path are shown in Figure 6. The blue line in the distance matrix grayscale is the optimal matching path.

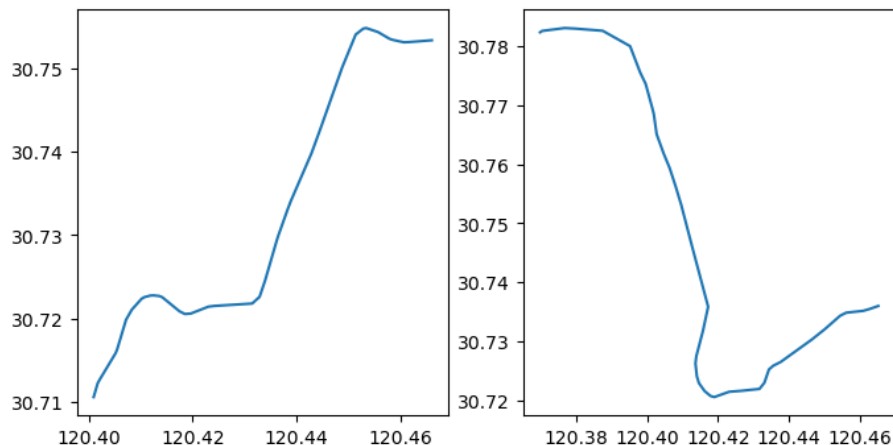

**Figure 5.** Two different ship sailing tracks.

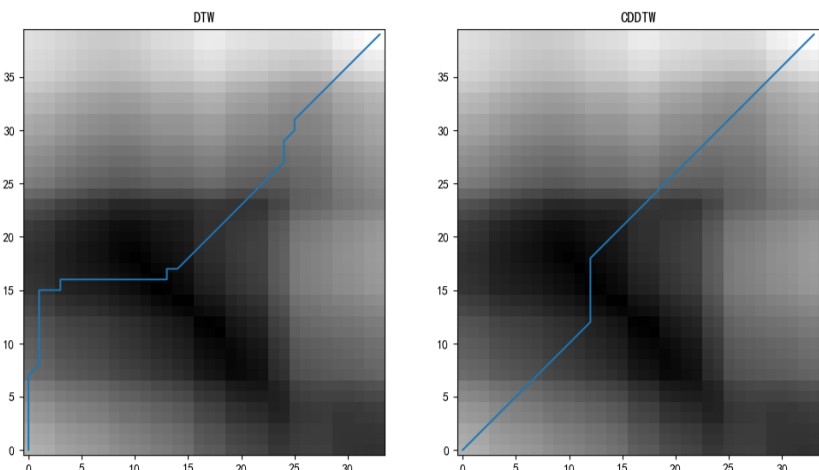

**Figure 6.** Gray similarity distance matrix and matching path of two trajectories before and after the improvement of the DTW algorithm.

3.2.2. Evaluation of the CDDTW Trajectory Clustering Algorithm

1. Evaluation indicators

The clustering result of ship trajectories can be judged by comparing the distance within the cluster after clustering. This requires calculating the average distance from the characteristic trajectory of the cluster to all other trajectories of the cluster within the same cluster [41]. Formula (2) is expressed as

$$S(C_i) = \frac{1}{n}\sum_{j!=i,j=0}^{n} dis(Ti, Lj), \qquad (2)$$

where $S(C_i)$ is the average distance within the cluster, $C_i$ is the cluster, $i$ = { 1, 2, 3,......, m, m is the total amount of clusters, $n$ is the total number of trajectories of cluster $C_i$, $Ti$ is the characteristic trajectory of cluster $C_i$, $Lj$ is the trajectory other than the characteristic trajectory in cluster $Lj$, $j$ = { 1, 2, 3,......, m}, and dis ($Ti$, $Lj$) is the distance between the trajectories $Ti$ and $Lj$. $S(C_i)$ is a relative value, which is related to the length of the trajectories within the cluster.

Some clusters are obtained in accordance with different trajectory clustering algorithms. Comparing the intra-cluster distance of the same cluster, the smaller the distance, the more compact the cluster.

2. Comparison of experimental results

We compare the clustering effect of four algorithms on ship trajectories. As shown in Table 4, we used Formula (2) to calculate the intra-cluster distance of the clusters obtained by the four algorithms. The cluster obtained through CDDTW has a smaller intra-cluster distance in most cases, which means these clusters are more compact. The table indicates that the clusters obtained through the CDDTW algorithm are closer.

**Table 4.** Comparison of intra-cluster distance of the same class clusters under different clustering algorithms.

| Algorithm | Intra-Cluster Distance | | | | | | |
|---|---|---|---|---|---|---|---|
| | Cluster 1 | Cluster 2 | Cluster 3 | Cluster 4 | Cluster 5 | Cluster 6 | Cluster 7 |
| DTW | 437 | 798 | 632 | 750 | 339 | 568 | 466 |
| CDDTW | 415 | 469 | 458 | 322 | 343 | 542 | 430 |
| HDBSCAN | 462 | 543 | 463 | 512 | 351 | 497 | 393 |
| OPTICS | 431 | 586 | 538 | 487 | 387 | 572 | 449 |

### 3.3. Construction of the RSTPM Based on LSTM

On the basis of the clustering results obtained in Section 3.2, different types of trajectory clusters should be trained separately to obtain their respective prediction models, which can improve the accuracy of prediction [24]. The RSTPM is constructed based on LSTM. Figure 7 shows the structure of the model.

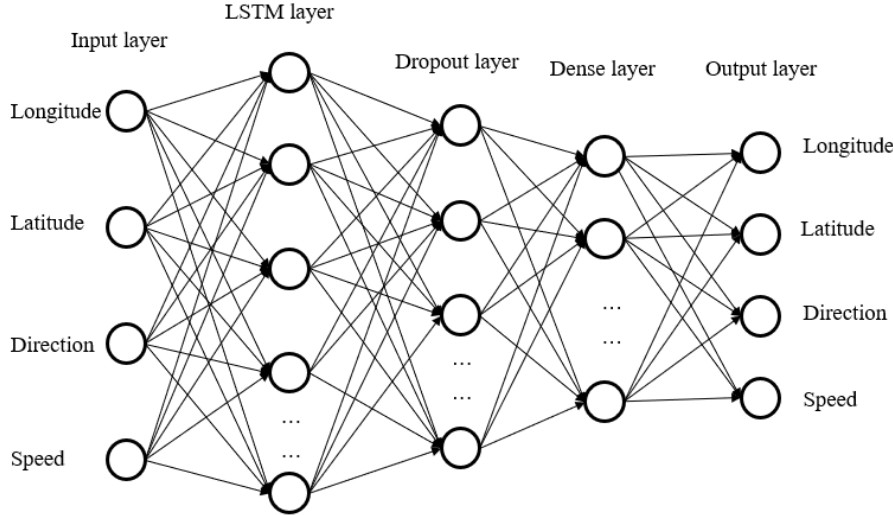

**Figure 7.** Structure of the RSTPM.

The navigation state of the ship at the first T continuous time is used by the RSTPM to predict the behavior of the ship at the future moment t. Although the model can only predict the navigation state of one moment at a time, we use the time window sliding method to forecast the state of the ship at multiple moments in the subsequent time. The Formula (3) for the single prediction of ships is

$$S(t+1) = f(\{S(t), S(t-1), \ldots, S(t-T+1)\}),  \tag{3}$$

$$S(t) = \{LON, LAT, SOG, COG\},  \tag{4}$$

where $S(t)$ represents the state of the ship at the current moment, $S(t+1)$ represents the state of the ship at the next moment, $f$ is the prediction model, and $T$ is the step size, indicating that $T$ historical data points form a set for the prediction of the ship's state at the next moment, where $LON, LAT, SOG, COG$ represent longitude, latitude, speed, and heading, respectively. The state $S(t)$ of the ship at each moment includes four characteristics: longitude, latitude, heading, and speed. Only Formula (3) needs to be looped when predicting the future multiple states of the ship, at which point Formula (3) becomes Formula (5):

$$S(t+2) = f\{S(t+1), S(t), \ldots, S(t-T+2)\}.  \tag{5}$$

Formula (5) shows that the data of the input model begin to include the previous prediction results because the prediction time exceeds $t+1$. Therefore, the prediction accuracy will keep declining as the prediction time grows.

We obtain real-time AIS data as the input of the model in two ways to realize the real-time prediction. One is to access data from AIS data service providers, and the other is to build an AIS data receiving system, then push the parsed AIS data continuously through the message queue. Next, the characteristic area to which the ship belongs is calculated in accordance with the current position data of the ship. Finally, the RSTPM can implement the prediction of ship trajectories in real-time. Within the scope of the study area, the model will continue to receive updated ship locations, and these data will be applied to cluster analysis to continuously optimize the prediction model.

## 4. Adaptability Evaluation of the RSTPM in Different River Types

We mainly study the applicability of the RSTPM on two different types of rivers. The first study area is the Wuhan section of the dendritic river system in the Yangtze River's middle reaches, which is a nonforked river section, as shown in Figure 8. The second study area is a partially reticulated river system in the northern part of the Zhejiang Province, where the Beijing–Hangzhou Canal intersects with the Dongzong Line. It is a multiforked river section, as shown in Figure 9. For the nonforked reach, we mainly focus on the problem of selecting model parameters and predicting accuracy under different river curvatures. For multiforked river sections, we focus on the multisegment trajectory prediction problem for ships navigating different bifurcated segments at channel bifurcations.

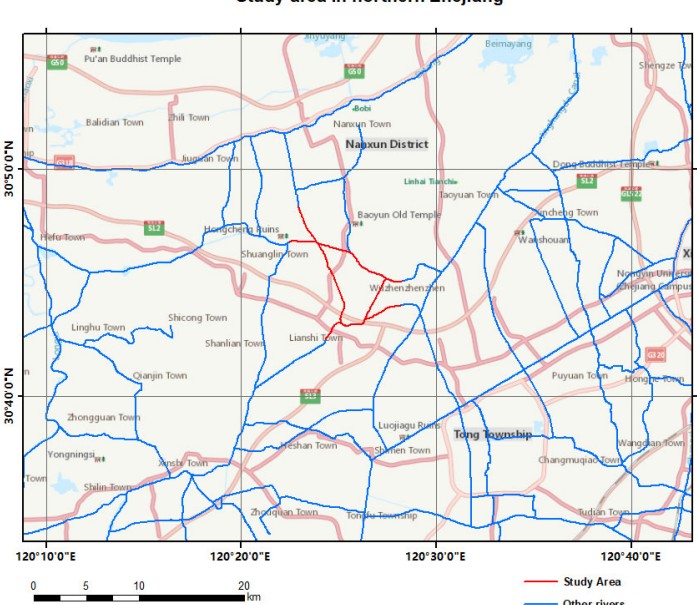

**Figure 8.** Nonforked section of the dendritic river system in the Yangtze River's middle reaches in Wuhan.

**Figure 9.** Some multiforked river sections of reticulated river systems in northern Zhejiang Province.

### 4.1. Nonforked River Sections

For nonforked river sections, we mainly focus on the impact of various river sinuosities on the selection of model parameters and prediction accuracy. The river sinuosity is calculated using Equation (6) [42] as:

$$C = \frac{L}{L'}, \tag{6}$$

where $L$ is the length of the river section, and $L'$ is the length of the connection line from the beginning to the end of the river section. We then applied the CDDTW to cluster ship trajectories on the waters of the first study area and obtained a total of 28 class clusters. We selected six different types of clusters to explore, using the area where they were located as the study area. The river segments in each study area should have different river sinuosities, as shown in Figure 10.

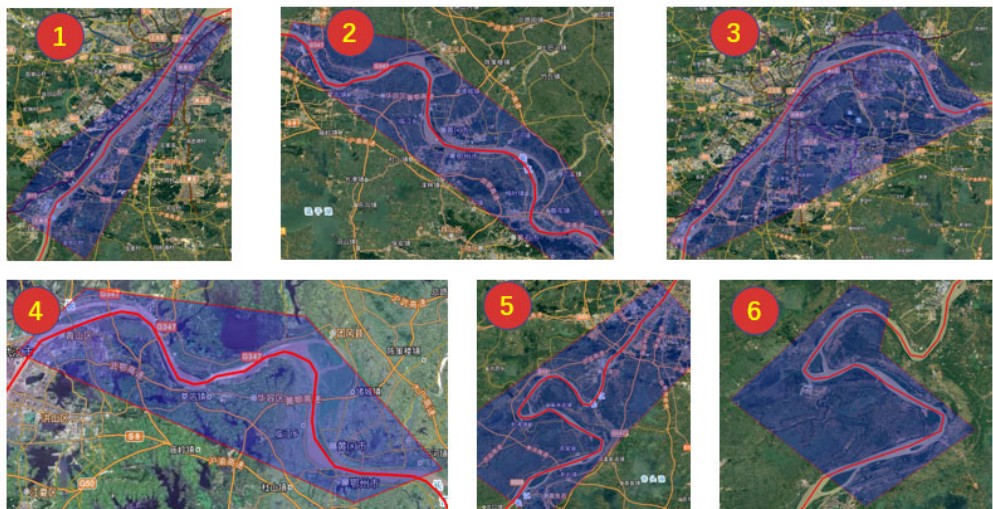

**Figure 10.** Study area and numbering.

We take the second study area as an example to investigate the effects of batch size, the amount of neuron nodes in the hidden layer in LSTM, and the step size of training data on the final prediction accuracy of the RSTPM under different river sinuosities. The parameter settings in the other areas are the same as in this area.

#### 4.1.1. Effect of Batch Size on the Prediction Model

Taking the second study region as an example, we set different values for batch size and trained them for 500 epochs. Mean squared error function is applied to calculate the error between the predicted value and the true value. Figure 11 shows that the convergence speed of the network on the training set is significantly higher than that with batch sizes of 16 and 32 when using batch sizes of 64, 128, and 256. On the validation set, batch sizes of 32 and 128 can achieve better results. Therefore, setting the batch size to 128 can better balance the error in the training set and validation set.

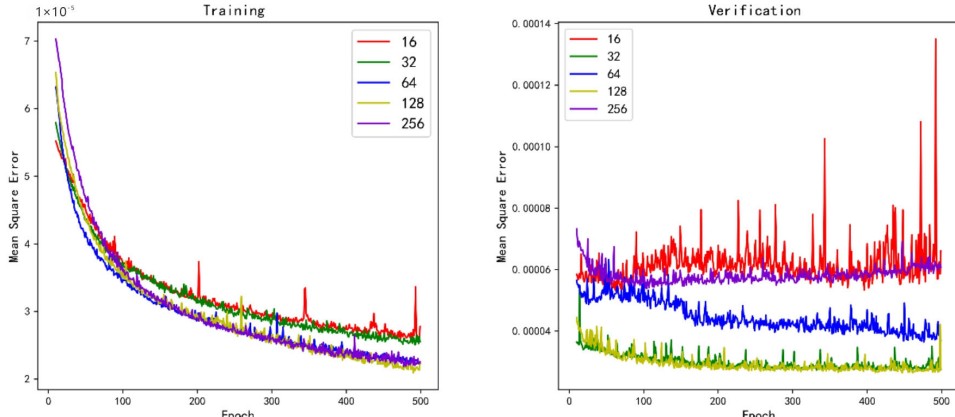

**Figure 11.** Mean squared error for different batch sizes in the training set and validation set.

### 4.1.2. Effect of the Amount of Hidden Layer Neuron Nodes in LSTM on the Prediction Model

As shown in Figure 12, when the number of hidden layer neuron nodes of LSTM is more than 108, the loss function on the training set no longer decreases significantly, whereas the validation and test sets continue to decrease significantly. Considering that the training time of the model becomes longer as the number of hidden layer neuron nodes increases, the number of LSTM hidden layer neuron nodes is taken as 128 by combining the error, accuracy, and training time of the model.

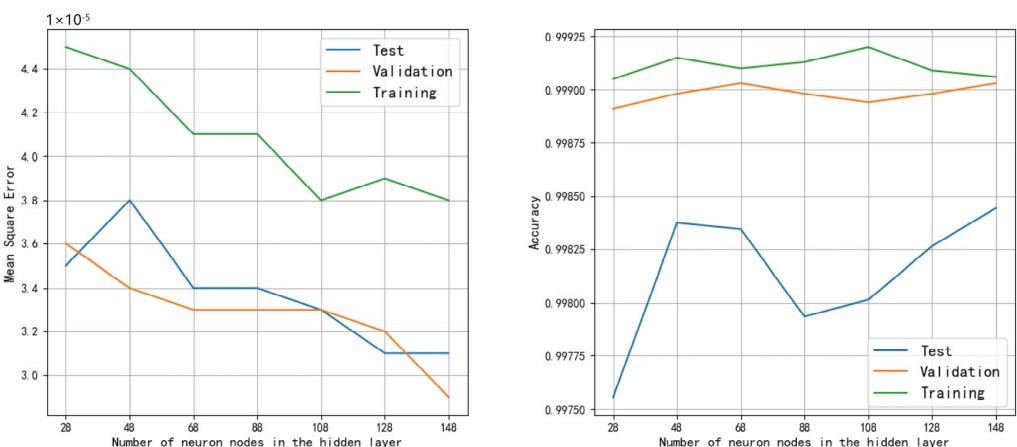

**Figure 12.** Variation in mean squared error and accuracy for the model with the number of neurons.

### 4.1.3. Effect of Step Size on the Prediction Model

As shown in Figure 13, the mean squared error of the training set and test set reaches its minimum value when the step size is 12, and the training set obtains the best accuracy, reaching 99.9%. This level of accuracy fully satisfies the requirement of accuracy, so the step size is set to 12 for the study area.

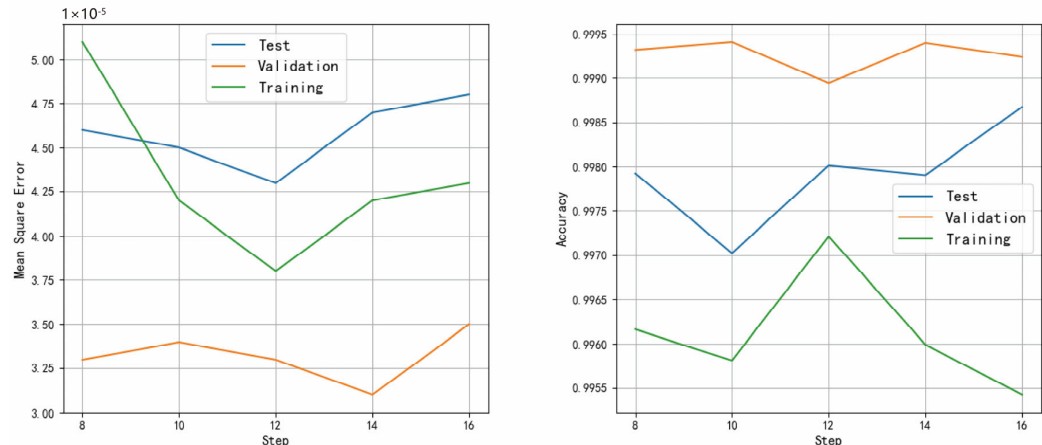

**Figure 13.** Variation of mean squared error and accuracy of the model with step size.

For the next five different river segments, we used the same method to determine the parameter selection of the prediction model under different river sinuosities. In accordance with the above analysis, the value of batch size is taken as 128 in all situations. After extensive experiments, the relationship between model parameters and accuracy for different river sinuosities is shown in Table 5. As the river sinuosity increases, the model needs fewer steps and more neurons to achieve higher accuracy.

**Table 5.** Parameter values of the RSTPM with different river bending coefficients.

| Model Parameters | Study Area | | | | | |
|---|---|---|---|---|---|---|
| | 1 | 2 | 3 | 4 | 5 | 6 |
| River bending coefficient (m/m) | 1.017 | 1.269 | 1.287 | 1.343 | 1.746 | 2.493 |
| Number of hidden layer neurons (pcs) | 88 | 108 | 128 | 108 | 168 | 188 |
| Training data step (step) | 14 | 12 | 12 | 12 | 10 | 8 |
| Predicted average accuracy (m) | 35 | 20 | 40 | 20 | 57 | 24 |

### 4.2. Multifork River Section

Predicting ship trajectories on multifork river sections is more difficult compared to nonfork river sections. Although most ships follow a consistent trajectory, some may choose different routes at channel bifurcations. Therefore, in addition to predicting the ship's position, we need to consider whether there will be multiple sailing routes after the ship travels to the bifurcation of the channel. The study area is shown in Figure 14, and the clustering results of ship trajectories are shown in Figure 15. The left side of the figure shows the clustering results with nine class clusters; the two obscured clusters in the left figure are shown on the right side.

On the basis of the river segments where the nine types of ship trajectory class clusters are located in Figure 14, the river segment sinuosities are calculated using Equation (6). We also select the RSTPM parameters for different segments based on the results in Table 6.

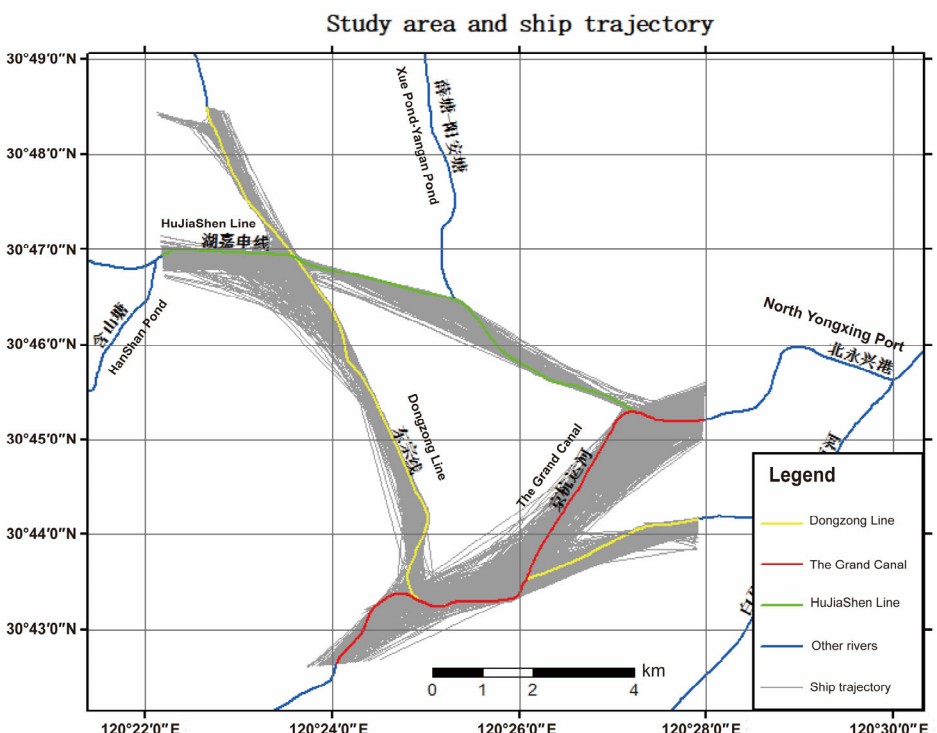

**Figure 14.** Channel names and ship tracks in the study area.

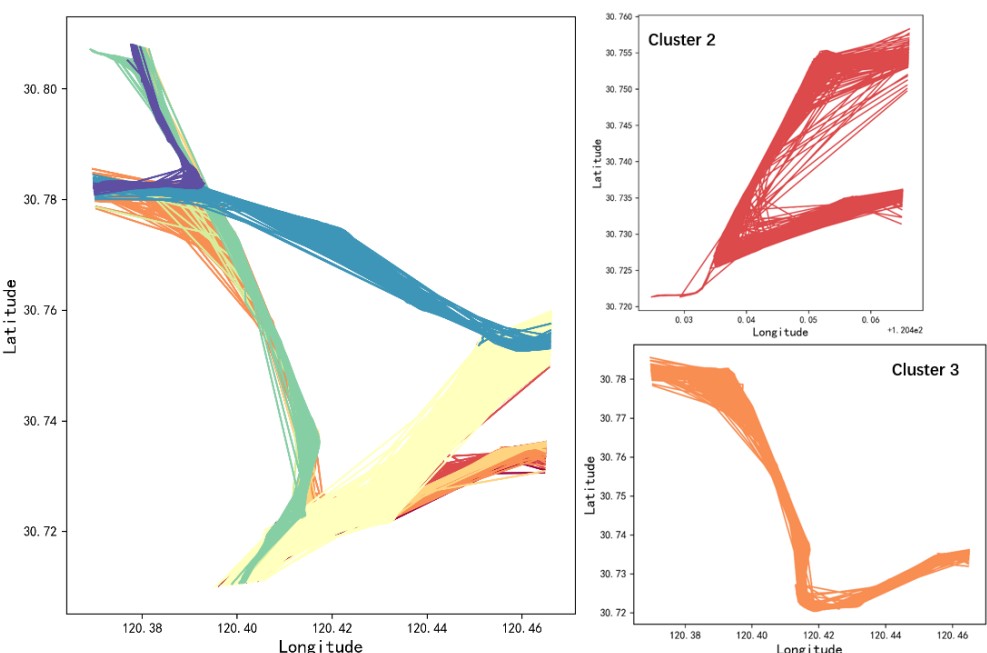

**Figure 15.** Vessel trajectory clustering results.

**Table 6.** Bending coefficients of river sections belonging to different clusters and the parameter values of RSTPM.

| Model Parameters | Cluster | | | | | | | | |
|---|---|---|---|---|---|---|---|---|---|
| | 1 | 2 | 3 | 4 | 5 | 6 | 7 | 8 | 9 |
| River bending coefficient (m/m) | 1.078 | 4.509 | 1.368 | 1.342 | 1.930 | 1.156 | 1.384 | 1.152 | 1.038 |
| Number of hidden layer neurons (pcs) | 128 | 188 | 108 | 108 | 168 | 168 | 108 | 108 | 128 |
| Training data step (step) | 14 | 8 | 12 | 12 | 10 | 10 | 12 | 12 | 14 |
| Predicted average accuracy (m) | 14 | 52 | 16 | 15 | 30 | 14 | 21 | 24 | 16 |

## 5. Experiments on the RSTPM

### 5.1. Analysis of Results in the Nonfork River Section

Taking the second one in Figure 10 as the study area, we selected a trajectory from the test set that was considered normal. The specific prediction results are shown in Figure 16, revealing an average prediction error of 18.4 m that does not exceed 60 m. Our prediction accuracy is significantly higher compared to the average prediction error of 60.3 m [16] and the average prediction error of 123 m [43] in other studies.

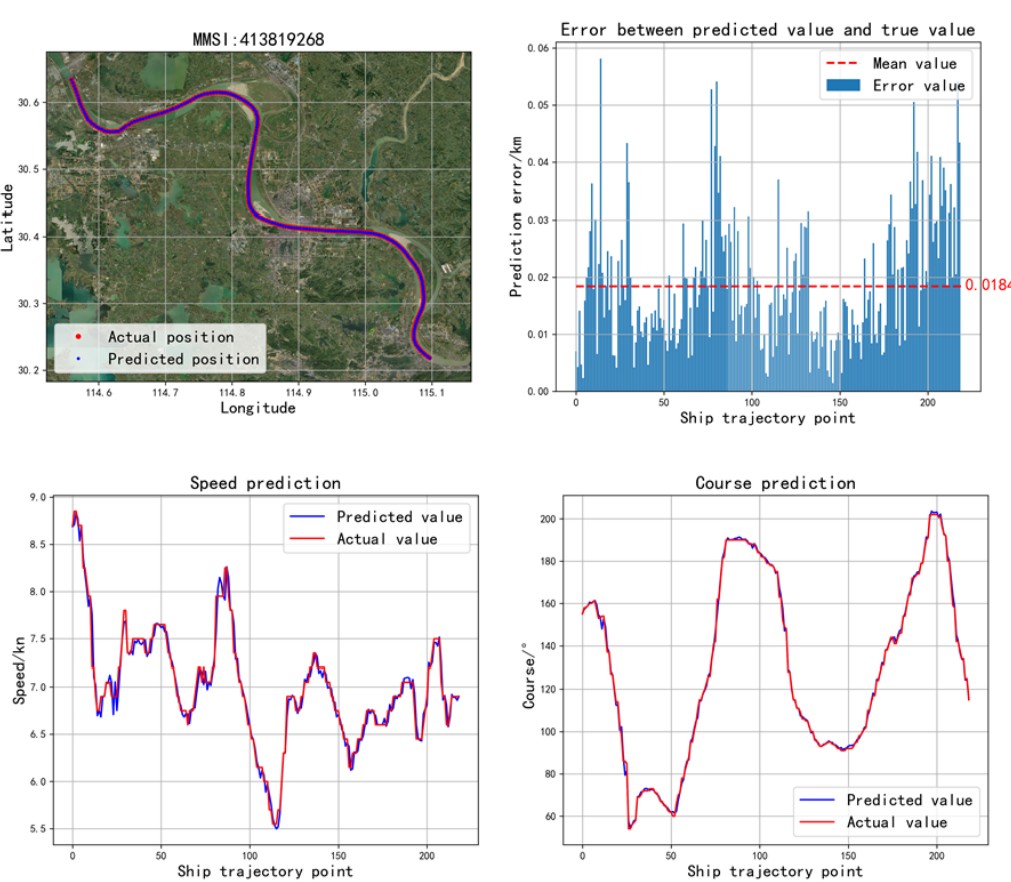

**Figure 16.** Normal trajectory prediction results and prediction error.

The two models of RSTPM and GRU for predicting 30 min on the test trajectory (MMSI:413819268) are shown in Figure 17. From the two figures, the distance error of the model gradually increases as the prediction time increases, while the error becomes lower in some positions as time increases.

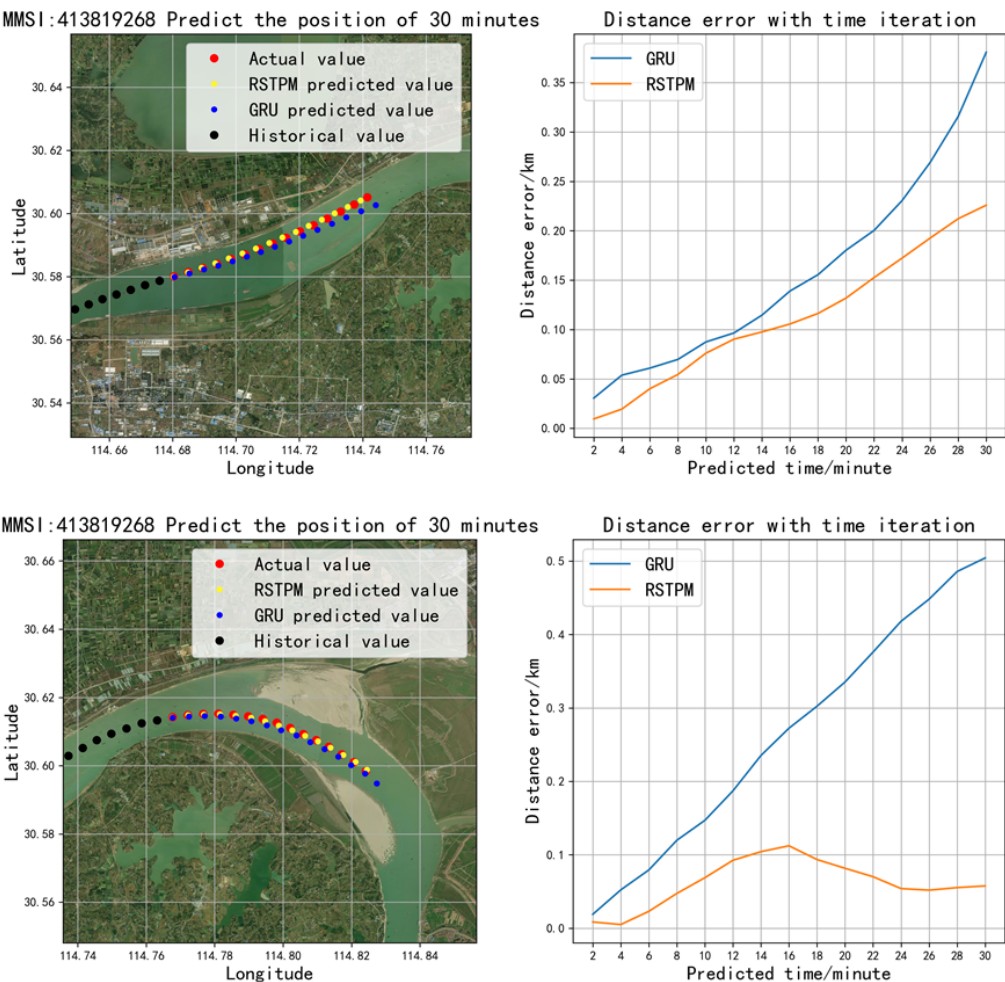

**Figure 17.** Prediction of trajectory within 30 min periods in different positions.

For predicting the trajectory of the ship (MMSI:413819268) across the whole trajectory, we made multiple predictions from the first position to the last position of the ship, and each prediction spanned 30 min. Figure 18 shows the average prediction error within 30 min across the entire trajectory. It demonstrates that the RSTPM outperforms the GRU model in terms of mean and maximum values, and that the mean prediction error of the RSTPM does not exceed 100 m within 12 min.

To confirm the model's stability, four trajectories were arbitrarily chosen from the test set for further testing. Table 7 shows the prediction errors for two different models. It indicates that the prediction distance error of the RSTPM is better than that of the GRU model, and the maximum value of prediction error of the RSTPM is less than 90 m. The average prediction error of heading and speed of the RSTPM is also less than that of the GRU model. The maximum prediction error for the former speed does not exceed 0.8 knots, while the maximum prediction error for heading is approximately 12°. Therefore, it indicates that the RSTPM is superior to the general trajectory prediction model and can better meet the demand of traffic supervision when the ship travels on the waterway surface.

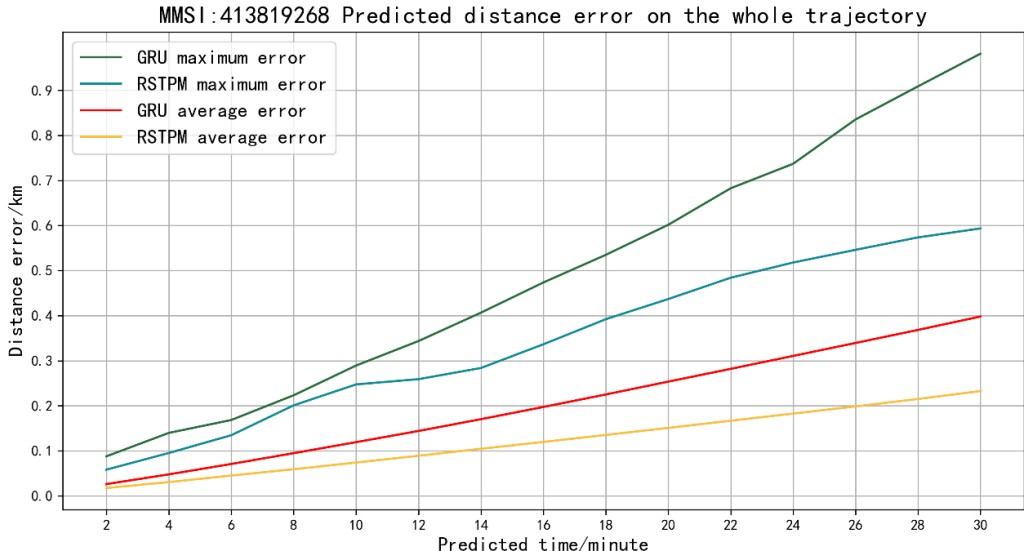

**Figure 18.** The average prediction error and the maximum error for the 30 min period across the entire trajectory.

**Table 7.** Comparison of prediction errors for different trajectories.

| MMSI | Models | Distance Error (km) | | Speed Error (km) | | Heading Error (Degree) | |
|---|---|---|---|---|---|---|---|
| | | Max | Mean | Max | Mean | Max | Mean |
| 413827414 | GRU | 0.0922 | 0.0269 | 0.331 | 0.069 | 6.210 | 1.130 |
| | RSTPM | 0.0884 | 0.0204 | 0.360 | 0.063 | 5.664 | 1.022 |
| 413812757 | GRU | 0.0734 | 0.0285 | 0.526 | 0.082 | 11.524 | 1.261 |
| | RSTPM | 0.0758 | 0.0208 | 0.537 | 0.073 | 12.157 | 1.121 |
| 413812608 | GRU | 0.0692 | 0.0266 | 0.997 | 0.078 | 7.214 | 1.267 |
| | RSTPM | 0.0758 | 0.0211 | 0.793 | 0.072 | 7.234 | 1.188 |
| 413819268 | GRU | 0.0877 | 0.0266 | 0.531 | 0.069 | 9.041 | 1.126 |
| | RSTPM | 0.0581 | 0.0184 | 0.551 | 0.066 | 7.010 | 0.997 |

### 5.2. Analysis of Results in Multifork River Sections

In this paragraph, we carry out some tests on ship trajectories in the multifork area selected in Section 4.2. As shown in Figure 19, one four-fork and three three-forks are observed in the experimental area (red dashed circles in the figure). When a ship travels to a three-fork area, it may travel to any of the other two segments. Therefore, the ship's position on two different segments needs to be predicted simultaneously before the ship arrives at the fork.

We use the same method as in Section 5.1 to verify the accuracy of the RSTPM and GRU models on trajectory prediction. Then we compare the prediction errors of four random trajectories from the test set in the multifork area, as shown in Table 8. It indicates that the RSTPM has lower errors than the GRU model in most cases.

We chose a ship from the test set that had traveled in multiple directions at a multifork river for testing. As shown in Figure 20, the ship (MMSI:413760772) has a total of five historical trajectories. Four of the historical trajectories of the ship belong to cluster 1 and one belongs to cluster 6, indicating an 80% probability of the ship passing through the channel where cluster 1 is located and a 20% probability of it passing through the channel where cluster 6 is located when it reaches the multifork.

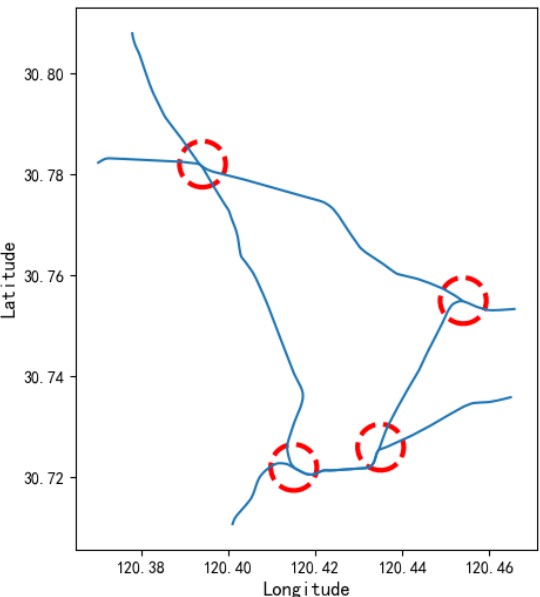

**Figure 19.** Location of multifork river in the study area.

**Table 8.** Prediction errors of different trajectories.

| MMSI | Models | Distance Error (km) | | Speed Error (km) | | Heading Error (Degree) | |
|---|---|---|---|---|---|---|---|
| | | Max | Mean | Max | Mean | Max | Mean |
| 413823145 | GRU | 0.0442 | 0.0171 | 0.362 | 0.098 | 5.194 | 1.433 |
| | RSTPM | 0.0362 | 0.0144 | 0.322 | 0.096 | 4.417 | 1.158 |
| 413791454 | GRU | 0.0354 | 0.0149 | 0.293 | 0.068 | 10.852 | 2.135 |
| | RSTPM | 0.0354 | 0.0131 | 0.286 | 0.069 | 7.434 | 1.847 |
| 413981121 | GRU | 0.0382 | 0.0154 | 0.507 | 0.110 | 5.003 | 1.577 |
| | RSTPM | 0.0397 | 0.0147 | 0.555 | 0.111 | 7.360 | 1.570 |
| 413977486 | GRU | 0.0300 | 0.0163 | 0.213 | 0.063 | 6.292 | 2.110 |
| | RSTPM | 0.0285 | 0.0138 | 0.210 | 0.064 | 6.794 | 1.627 |

| | traj_id | cluster | geometry | mmsi |
|---|---|---|---|---|
| 2701 | 413760772_2021-10-09 09:06:13_0_0 | 1 | LINESTRING (120.40091 30.71059, 120.40112 30.... | 413760772 |
| 2705 | 413760772_2021-11-23 15:49:09_0_0 | 1 | LINESTRING (120.40095 30.71059, 120.40103 30.... | 413760772 |
| 2715 | 413760772_2021-08-10 18:10:24_0_0 | 1 | LINESTRING (120.40093 30.71059, 120.40124 30.... | 413760772 |
| 2722 | 413760772_2021-08-23 16:31:49_0_0 | 1 | LINESTRING (120.40084 30.71058, 120.40097 30.... | 413760772 |
| 7035 | 413760772_2021-08-28 15:26:44_0_0 | 6 | LINESTRING (120.40095 30.71059, 120.40110 30.... | 413760772 |

**Figure 20.** Ship history track.

As shown in Figure 21, we attempted to predict the ship's trajectory on two different segments prior to its arrival at the fork. According to the prediction accuracy of the two models, it is hard to determine exactly which river segment the ship will choose. However, we can obtain the probability of the ship traveling on two segments based on its historical trajectory data. The figure shows that the prediction result of model 6 at 2 min deviates significantly from the centerline of the channel, and the prediction error is more than 10 m, whereas the prediction error of model 1 is less than 10 m. Therefore, the ship will pass through the section where the class cluster 1 is located. The final prediction result for the

ship's trajectory is shown in Figure 22. The results prove the applicability of the developed model in the multifork river sections.

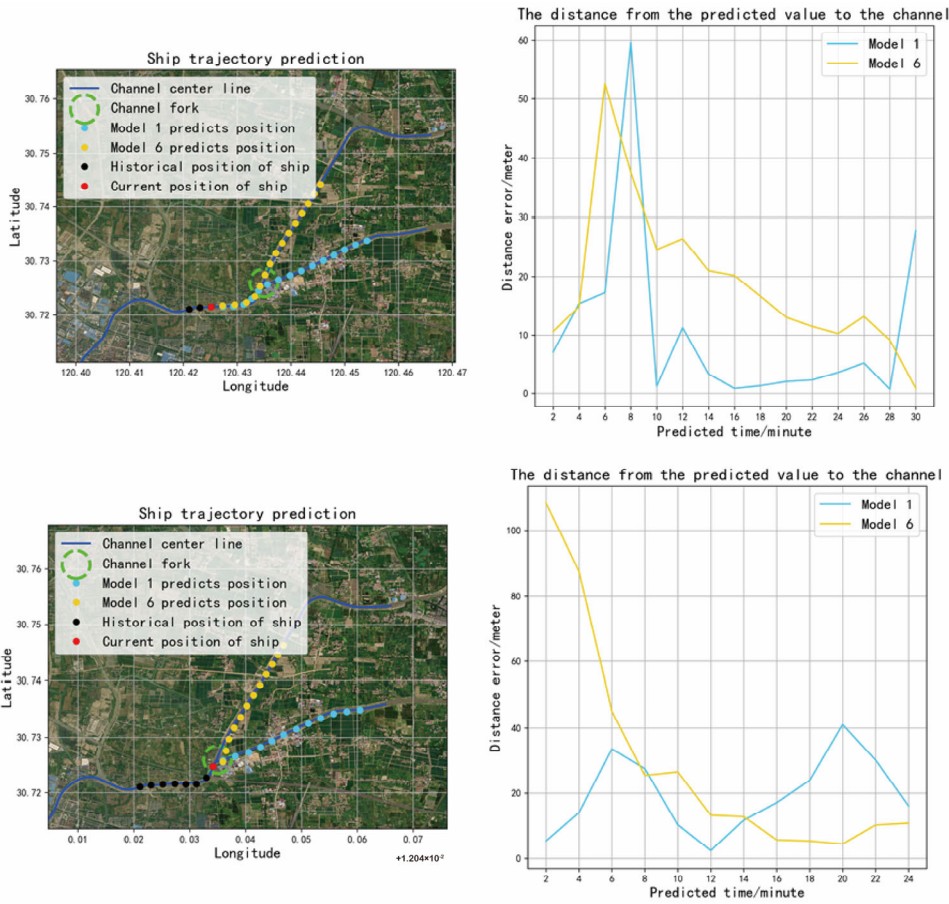

**Figure 21.** Predicted trajectory on multiple segments when the ship reaches the bifurcation.

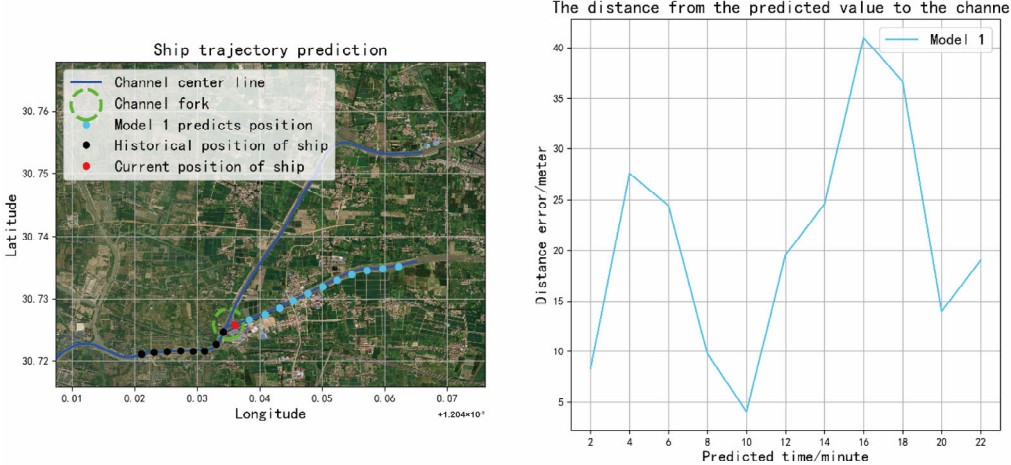

**Figure 22.** Predicted trajectory of the ship after passing the bifurcation.

## 6. Conclusions

In this study, we focused on the Wuhan section of the dendritic river system in the Yangtze River's middle reaches and a partial reticulated river system in the northern part of the Zhejiang Province. Using AIS track data, we proposed a new ship track clustering method called CDDTW, which improves the clustering efficiency and quality of ship tracks and effectively extracts regional navigation characteristics of ships. We then constructed

an RSTPM based on the trajectory clustering results and an RNN algorithm. The results of the experiment suggest that the proposed model can predict ship trajectories on inland waterways with an accuracy of approximately 20 m, characterized by high detection accuracy, less setting parameters, and ease of implementation.

We hope our research can provide theoretical and technical support to the water traffic management department, which is of practical importance in ensuring that ships can navigate safely on the waterway and improving the department's ability to manage the ships. In the future, we strive to enhance the model's prediction accuracy by increasing the input dimensions, such as ship length and ship type.

**Author Contributions:** Conceptualization, Daping Xi and Wenping Jiang; methodology, Yuhao Feng; software, Xini Hu; validation, Chuyuan Wang, Yuhao Feng, and Xini Hu; formal analysis, Daping Xi; investigation, Wenping Jiang and Xini Hu; resources, Daping Xi; data curation, Yuhao Feng; writing—original draft preparation, Daping Xi; writing—review and editing, Yuhao Feng, Wenping Jiang, and Xini Hu; visualization, Chuyuan Wang; supervision, Xini Hu; project administration, Nai Yang; funding acquisition, Nai Yang. All authors have read and agreed to the published version of the manuscript.

**Funding:** This research was funded by National Natural Science Foundation of China, grant number 42171438 and 41371428.

**Data Availability Statement:** The code used to generate the results of this study is publicly available at https://doi.org/10.6084/m9.figshare.23501766 (accessed on 12 June 2023). URI: https://figshare.com/articles/software/realtime_ship_trajectory_prediction_model/23501766 (accessed on 12 June 2023).

**Acknowledgments:** The author would like to thank the teachers and students at China University of Geosciences (Wuhan) for their suggestions and help. We are grateful to the anonymous reviewers and the editors of the journal for their important contributions in improving the article for publication.

**Conflicts of Interest:** The authors declare no conflict of interest.

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
