# Peer review of "Construction of a Real-Time Ship Trajectory Prediction Model Based on Ship Automatic Identification System Data"

_ijgi, doi:10.3390/ijgi12120502_

Round 1
Reviewer 1 Report
Comments and Suggestions for Authors
Based on historical AIS trajectory data, a ship trajectory prediction method (called RSTPM) is proposed in this paper, which has some significance for ship accident warnings and other applications. The presented method first divides the AIS trajectories into clusters and then builds the ship trajectory prediction model based on LSTM for each trajectory cluster, which can improve the prediction accuracy. Finally, experimental analysis of AIS trajectory data from continuous river sections and multiforked river sections were carried out to verify the effectiveness of the proposed method. Compared with the existing studies, this method proposes an improved ship trajectory clustering algorithm that integrates DBSCAN and DTW and combines the characteristics of different trajectory clustering to build prediction models. In general, this paper has specific innovations. However, some major issues still need to be explained clearly.
1) The introduction of AIS trajectory data preprocessing is too brief, and it is suggested to supplement the description of critical steps, such as the interpolation of missing data in trajectories.
2) The description of the trajectory clustering algorithm CDDTW is not very clear, especially the improvement of the DBSCAN and the DTW algorithms is not clearly explained, and the critical steps of the clustering algorithm were not described. At the same time, how to set the parameters of the trajectory clustering algorithm, for example, density threshold, similarity threshold, etc., is not explained.
3) In Line 189, what does the DTW algorithm mean? There is no reference or explanation of the DTW algorithm. In addition to intra-cluster compactness, inter-cluster separation should also be considered in evaluating the quality of clustering results. DBCV, Davide-Bouldin Index, CDbw Index, Silhouette Coefficient, and other clustering evaluation indexes are recommended.
4) According to the description, the presented method first partitions the trajectories into different clusters and then builds different prediction models for different trajectory clusters. In practical application, how can to cluster the real-time trajectory data of a ship? How do we build a prediction model based on the real-time trajectory data of the ship? If it is necessary to cluster the trajectories according to the historical track data of the ship, how will the trajectory data of other ships be obtained? Are the ship trajectories within a particular regional area for a period of time used for the clustering analysis and prediction? If so, a supplementary explanation is needed.
5) The overview framework of this paper is similar to that of Suo et al.[27], so it is suggested to do a comparative experiment with this method to illustrate the superiority of the proposed method.
6) The experiment section is disorganized, making it difficult for the readers to understand. At the same time, the statistics of the used trajectory data and the setting of algorithm parameters used in the experiment were not clear.
Other minor issues:
1) What is the meaning of the symbols in formula (4), such as SOG, COG?
2) Please check the symbols and text in the figures, e.g., Figure 12.
Comments on the Quality of English Language
Minor editing of English language required.
Reviewer 2 Report
Comments and Suggestions for Authors
The paper presents an interesting methodology for real time trajectory prediction for ships. I like the approach and the results and discussion.
Comments on the Quality of English LanguageI recommend rephrasing most of the abstract and introduction.
Reviewer 3 Report
Comments and Suggestions for Authors
To reduce the occurrence of ship accidents, this paper proposes a ship tracking clustering method called CDDTW to extract the navigation characteristics of ships. Furthermore, a real-time ship trajectory prediction model (RSTPM) is constructed to provide real-time prediction of ship trajectories. The study selected two river section as experimental areas, and found that the proposed approach has a higher prediction accuracy than traditional trajectory prediction models. However, there are several issues that need to be revised:
1.The names of sections 3.2.1 and 3.2.2 are identical.
2. The experiment in Table 3 only compares CDDTW and DTW algorithms, which is incomplete without including experiments with other clustering methods such as DBSCAN.
3. The RSTPM, which was created for real-time predictions, should provide empirical data to support the model's prediction time.
4. Based on Figure 6, it can be inferred that RSTPM is essentially an application of LSTM, lacking substantial innovation.
5. The basis for selecting only 6 out of 28 cluster types in Section 4.1 and 9 cluster types in Section 4.2 should should be explained and justified.
6. The reasons for only comparing GRU and RSTPM in the experiment in Section 5 should be explicitly stated.
Round 2
Reviewer 1 Report
Comments and Suggestions for Authors
The author has carefully revised the questions raised by the reviewer and responded one by one. As for the revised version, I think the quality has been greatly improved. I don't have any other questions.
Author Response
We appreciate your recognition of the revised version of the paper, and we will continue to improve and optimize our research in the future. Thank you very much for your comments and suggestions.

Reviewer 3 Report
Comments and Suggestions for Authors
The author has worked diligently during the revision process and provided additional explanations. However, there are still some issues that have not been clearly explained and have not been improved.
1. The experiments with clustering algorithms still cannot fully demonstrate the optimization effectiveness of existing improvements.
As stated in the paper, “We introduce Ball-Tree algorithm into DBSCAN algorithm to accelerate the query speed of neighborhood points”. However, how to prove that this algorithm can indeed accelerate query speed is not explained clearly.
Also, as stated in the cover letter, “In this paper, the purpose of the trajectory clustering algorithm is to better construct the ship trajectory prediction model, so we do not focus on the comparison and optimization of more clustering algorithms, but in the subsequent experiments, we will take these as the key factors to further improve the accuracy of the model”. However, this does not explain why other clustering algorithms cannot be compared. It seems that the improvement of the DBSCAN algorithm has not been reflected in the clustering comparison experiment. Since clustering algorithms are used to improve ship trajectory prediction models, it would be worthwhile to compare with more clustering algorithms.
2. Response 3 to comments 3 cannot be used as a reason for not providing empirical data on the model's prediction time. Empirical data should be provided to prove that this algorithm can provide real-time prediction.
3. The paper only compared GRU and RSTPM in the experiment in Section 5. Suo et al. [27] used a GRU model to predict ship trajectories. As suggested by Reviewer 1, it is more appropriate to conduct comparative experiments with RSTPM and Suo et al. [27]. However, it seems that there were no corresponding comparisons in the revised version of the paper.
Round 3
Reviewer 3 Report
Comments and Suggestions for Authors
The revised paper has been revised as much as possible and there is unlikely to be further improvement by returning to the modifications. I agree to accept the paper as it is.